

# Review on Mineral Characterization of Precambrian Charnockites-using PIXE Technique

Avupati Venkata Surya Satyanarayana[1], Mokka Jagannadharao[2], Kemburu Chandra Mouli[3], Kollu Sai Satya Mounika[4]

1,3 Department of Engineering Physics, Andhra University, Visakhapatnam-530003, AP, India.

2,4 Department of Geology Andhra University, Visakhapatnam-530003, A.P, India.

Corresponding Author; Avupati Venkata Surya Satyanarayana (E. Mail; *savs.viit@gmail.com*)

*Abstract:*

Particle Induced X-ray Emission (PIXE) has been applied to an analytical tool for long range of major, minor, trace and REE elemental analysis in Precambrian Charnockites. PIXE is sensitive and non-destructive method for some elemental analysis in a variety of metamorphic Precambrian Charnockite rocks down to levels of a few parts per million and it is not valid for all remaining elements in the composition. The elements in the Precambrian Charnokite rock Cl, K, Ca, Ti, V, Cr, Mn, Fe, Ni, Cu, Zn, Se, Br, Rb, Sr, Y, Zr, Nb, Mo, Ru, Ag, Pb are identified without exact values by PIXE but the elements minor Be and F, major elements Na, Mg, Al, Si, P, Ba and traces of Sc, Ce, Co, Sn, W, Ge, Ga, Au, Th, U and REE not detected due to various reasons even though there present in the Charnockites, because of PIXE which is operation at 3MeV energy and characterization material of Charnockite mineral investigated. In mineral characterization of Charnockite rocks, elemental errors in concentration of the compositions explained by comparing with present other non nuclear technique and previous nuclear technique studies.

*Keywords*; PIXE Technique, Metamorphic, Precambrian, Charnockites, Multi elemental analysis, Mineral characterization, Review.

## 1. Introduction

The geological history of the earth is primarily divided into three periods. They are 1) Archian period 2) Proterozoic period 3) Phanerozoic period. The phonerozoic era is further divided into 1. Paleozoic (ancient life) 2. Mesozoic (middle life) and 3. Genozoic (recent life) periods. This classification is done on the basis of fossils found in each period. Again the first period of Paleozoic era is called Cambrian period and hence the time before the phanorozic era is also called pre-cambrian period which dates back to more than 2.5 billion years. As is well known the earth is conventionally divided into crust, mantle and core. The crust is further divided into three major categories continental, transitional and Oceanic. Geochemical and Petrologic investigations of Precambrian mafic Radha Krishna (2008) metamorphic rocks containing iron and magnesium) igneous rocks play an important role in establishing the evolution of the crust. Most Precambrian exposures Srivastava, and Ahmad (2008) (shields or cratons) contain metamorphic rocks and of rock were changed.



The charnockite series is a group of igneous rocks French, et. al., (2008) variably
metamorphosed. They are widely distributed and occupy an important place in the geology of
India is the main sources for creation of continental crust. Calc-alkaline rocks typically are
found above subduction zones, commonly in volcanic areas, and particularly on such area on
continental crust. A widely accepted theory of the development of earth's crust states that the
early earth would have had a proto crust formed from ultra mafic and felsic layers.

The samples analysed using PIXE in the present study are collected in a very
interesting context. Geologically the Visakhapatnam city is characterized by rocks termed as
Eastern Ghats. The rocks are Precambrian age and comprise mainly Khondalites, Lepitynites,
Poryxene Granulites and Charnockites and all of them have undergone metamorphosis.
Among these, charnockites are termed as upper mantle basic igneous rocks and are emplaced
into proto crustal rocks during Precambrian times. In this way the charnockites sometimes
may contain the relict bodies of earlier crustal rocks (proto crust).The proto crust is derived
from primitive oceans which are also called intra-cratonic sea water bodies.

The samples chosen for analysis are collected from the central portion of a charnockite
hill Rao, and Babu, (2008) near Visakhapatnam airport during a demolition operation for
extension of the airport. A big lenticular mass of relict lithological body which is
compositionally and physically different from the host charnockite was observed in this
central portion of the hill. This body is believed to be the caught up body of the early crustal
layer (possibly proto crust).Such samples are rare. PIXE technique is chosen for the trace
elemental analysis Kullerud, and Steffen, (1979) of these rare samples as it is a highly
sensitive and non destructive method for the simultaneous multi elemental analysis. Elements
present in ppm levels can be detected efficiently with this technique.

The experimental work was carried out using the 3Mev particle accelerator facility at
the institute of physics, Bhubaneswar. The characteristic X-rays were detected with Si (Li)
detector. The data analysis was carried out and concentrations of various elements detected
were determined using GUPIX software and different elements were detected in varied
concentrations. On the basis of the concentrations of these different trace elements obtained
using PIXE Technique, a geochemical analysis of the rock samples was performed and
interpreted for the genetic significance substantiating the information from previous
literature.

**2. Experimental Details and Data Analysis.**

Some of the trace elements are present in minute amounts in geological samples. Earlier
it was very difficult to measure their precise concentrations because of non availability of
sophisticated analytical methods. They were therefore described as occurring in traces, hence
the term `trace element'. With the invention of many modern analytical techniques Sie, et. al.,
(1989) like Atomic Absorption Spectrometry (AAS) Instrumental Neutron Activation
Analysis (INAA), Rutherford Back Scattering (RBS), X-Ray Fluorescence (XRF),Energy
Dispersive X-Ray Fluorescence (EDXRF), Auger Electron Spectroscopy (AES), Particle
Induced Gamma Ray Emission (PIGE),Particle Induced X-Ray Emission (PIXE),
Wavelength Dispersive X-Ray Fluorescence (WDXRF) etc. It has become possible to
estimate the concentrations of trace elements in ppm and ppb levels. These analytical
techniques have the capability to measure all the trace elements present even in the smallest
geological samples Malmqvist, (1987) with great precession and accuracy. The term trace is a
traditionally followed through it has become scientifically obsolete owing to the availability
improved techniques. Among all the afore-mentioned techniques, PIXE Technique has its
own advantages over the other techniques. From analytical point of view, techniques for the
identification of trace elements and evaluation of their concentrations are categorized into
destructive and non destructive techniques. Tangi, (1998) Chemical analysis and AAS are the
two well known methods under the former category. Generally these methods require large
amounts of sample and are tedious as they involve element-by-element analysis.
PIXE and XRF are the both the methods based on x-ray emission are have several
features in common. From sensitive point of view PIXE has certain superiority .Moreover the
bremsstrahlung produced in PIXE is a secondary effect and hence is also the principle
determinant of detection limits. The low bremsstrahlung in PIXE enables parts per million
sensitivities, superior to its sister techniques. Due to high sensitivity and multi elemental
analysis capability, PIXE has found applications in trace elemental analysis Luciana (1999)
of samples from almost every conceivable field of scientific or technical interest. Some of
these fields are Biomedicine, Environment, Archaeology, Material science, Forensic studies,
Industrial applications and Geology.
The present study is aimed at estimating the concentrations of different trace elements
in geological samples of Precambrian charnockite hill near Visakhapatnam airport using
particle induced X-ray emission (PIXE) technique. These experiments are carried out using
3MV pelletron accelerator facility at the Institute of physics, Bhubaneswar. Protons with
3Mev energy are used to excite the samples. The samples are mounted on an Aluminium
target holder (a ladder arrangement).Then the target holder is inserted into the scattering
chamber and the irradiation is carried out in vacuum conditions. A collimated proton beam of
2 mm diameter is made to fall on to the sample. The beam current is kept at 20 nA. The
samples on the target holder which are to be exited or positioned in this scattering chamber at
an angle of $45^0$ with respect to the direction of the proton beam. The position of the sample
relative to the beam direction is adjusted properly by viewing through a window provided in
the scattering chamber. A high resolution Si (Li) detector (160 eV FWHM at 5.9 KeV
energy) is employed in the present experiments to record the X-ray spectrum. The detector is
placed at an angle of $90^0$ with respect to the beam direction. The output of the Si (Li) detector
is coupled to data acquisition system, which records the X-ray spectrum. The spectrum of
each sample is recorded for a sufficiently long time so as to ensure goods statistics. During
the irradiation of each sample the charge collected and the average beam  current is noted.
The Guelph PIXE (GUPIX) software package is used to analyse the spectra utilizing a
standard Marquardt non-linear least square fitting procedure. This package is provision to
identify different elements present in the sample and to estimate their relative intensities.
Using this GUPIX software package the X-ray intensities of different elements are converted
into the respective concentrations using a standardization technique involving fundamental
parameters, pre determined instrument constants and input parameters such as   solid angle,
charge collected etc. Comparing the concentrations of Yttrium obtained in the present work
with the known concentration of Yttrium added to the sample, the reliability of the input
parameters is checked. To assure the reliability of experimental system and other parameters,
in the same experimental conditions, the PIXE spectrum is recorded with NIST certified
reference material and the relative concentrations of different elements are estimated using
GUPIX software package. The relative concentrations of different elements thus obtained in
the present experiment for the above standard samples are compared with the certified
concentrations supplied by NIST. Good agreement Table-1 with in experimental uncertainties





is observed and this shows the reliability of the present experimental system and use of
GUPIX software package in the data analysis.
Table-1. PIXE spectrum is recorded with NIST certified reference material-Apple Leaves-
144 1515.

| Elements | Concentration (ppm) | |
|---|---|---|
| | **Certified values** | **Measured values** |
| K | 1.48±0.05 | 1.60±0.02 |
| Ca | 1.615±0.26 | 1.53±0.02 |
| Mn | 48.5±2.4 | 54.0±3.0 |
| Fe | 88.1±4.5 | 83.0±5.0 |
| Cu | 5.3±0.4 | 5.60±0.24 |
| Zn | 12.9±0.7 | 12.5±0.03 |
| Se | 0.06±0.01 | 0.05±0.009 |
| Rb | 9.3±1.0 | 10.2±1.50 |
| Pb | 0.54±0.08 | 0.47±0.02 |

This deals with the detailed explanation of the advantages of PIXE over other analytical
techniques, principle and technical minutiae of the PIXE technique, experimental setup,
sample preparation and data analysis.
**3. Results and discussions**
The PIXE spectrum fig; 1-7 of the geological samples G1 to G7 collected from the
interior of the charnockite rock recorded by Si (Li) detector. The concentrations in ppm of
these various elements in each sample were determined using the GUPIX software .These
concentrations are presented with errors in tables 2-13.

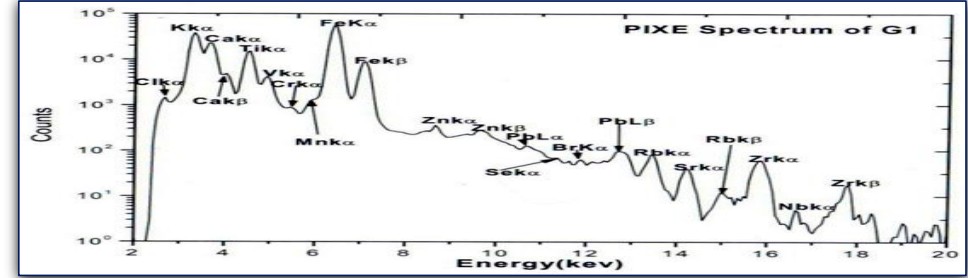

Figure-1; PIXE Spectrum of sample G1




Figure-2; PIXE Spectrum of sample G2

Figure-3; PIXE Spectrum of sample G3

Figure-4; PIXE Spectrum of sample G4

Figure-5; PIXE Spectrum of sample G5

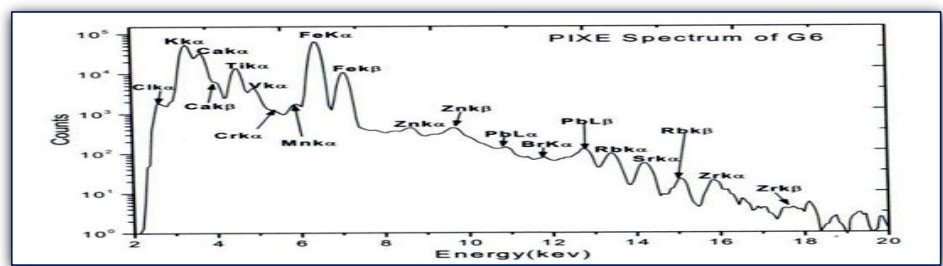

Figure-6; PIXE Spectrum of sample G6

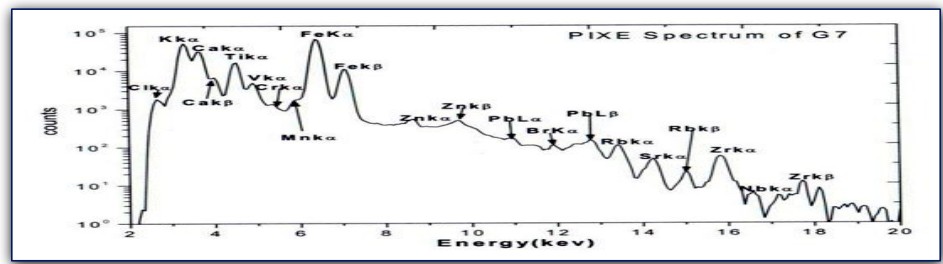

Figure-7; PIXE Spectrum of sample G7

The PIXE spectrum of the geological samples G1 to G7 collected from the interior of
the Charnockite rock recorded by Si (Li) detector. These concentrations are presented with
errors in Table-9. Another attempt is made to analyze the samples using an atomic absorption
photo spectrometer and same elements are reported using the method of AAS and the same
standard also employed for a method of AAS also. The data generated the AAS method has
been used to compare the PIXE results for its evaluation purpose. In this paragraph, each
element is considered in evaluating PIXE as per that element is concentrated. It is observed
that the results obtained by AAS are close which already published data on Charnockites in
various journals.

The possibility of increasing accuracy of PIXE in analysing samples of complex
matrix like Charnockite has been discussed and suggestions are made. So, the following
tables presents the elements which are close to elements having moderate errors and elements
which are highly erroneous not detected. Using this data, the Charnockite hill from where the
samples are collected has been attempted. to understand the chemical nature followed by
genetic implications. The reasons behind the poor performance of PIXE with respect to
certain elements have been tried to explain.





Table-2; G1 sample elements and
concentrations

| S.NO | Element | G1 |
|---|---|---|
| 1 | Cl | 394.1±16.5 |
| 2 | K | 4080±28.2 |
| 3 | Ca | 2229±25.9 |
| 4 | Ti | 1394±11.6 |
| 5 | V | 17.92±4.8 |
| 6 | Cr | 16.63±2.3 |
| 7 | Mn | 18.623.9± |
| 8 | Fe | 5200±20.3 |
| 9 | Ni | 10.96±3.7 |
| 10 | Cu | BDL |
| 11 | Zn | 9.147±3.3 |
| 12 | Se | 6.3±65±2.5 |
| 13 | Br | 4±2.2 |
| 14 | Rb | 48.87±6.0 |
| 15 | Sr | 38.5±5.4 |
| 16 | Y | BDL |
| 17 | Zr | 95.91±9.8 |
| 18 | Nb | 7.035±3.1 |
| 19 | Mo | BDL |
| 20 | Ru | BDL |
| 21 | Ag | BDL |
| 22 | Pb | 32.93±15.6 |


Table-3; G2 sample elements and
concentrations

| S.No | Element | G2 |
|---|---|---|
| 1 | Cl | 399.6±17.1 |
| 2 | K | 4187±25.1 |
| 3 | Ca | 2281±22.8 |
| 4 | Ti | 1271±9.3 |
| 5 | V | 23.69±4.1 |
| 6 | Cr | 38.53±2.1 |
| 7 | Mn | 34.71±3.7 |
| 8 | Fe | 6575±21.0 |
| 9 | Ni | 11.43±3.7 |
| 10 | Cu | BDL |
| 11 | Zn | 18.21±3.5 |
| 12 | Se | BDL |
| 13 | Br | 12.17±2.93 |
| 14 | Rb | 42.08±6.1 |
| 15 | Sr | 28.6±4.6 |
| 16 | Y | 12.64±4.5 |
| 17 | Zr | 20.86±6.0 |
| 18 | Nb | 9.812±3.9 |
| 19 | Mo | 24.34±6.3 |
| 20 | Ru | BDL |
| 21 | Ag | 12.36±9.0 |
| 22 | Pb | 38.35±17.3 |






Table-4; G3 sample elements and
concentrations

| S.NO | Element | G3 |
|------|---------|-----|
| 1 | Cl | 379±16.5 |
| 2 | K | 4148±26.1 |
| 3 | Ca | 2637±25.6 |
| 4 | Ti | 1109±9.3 |
| 5 | V | 7.85±4.0 |
| 6 | Cr | 15.16±2.1 |
| 7 | Mn | 34.31±3.7 |
| 8 | Fe | 5649±19.8 |
| 9 | Ni | 10.06±3.39 |
| 10 | Cu | 6.1±2.7 |
| 11 | Zn | 14.23±3.0 |
| 12 | Se | BDL |
| 13 | Br | 11.82±2.85 |
| 14 | Rb | 62.73±6.0 |
| 15 | Sr | 44.61±4.9 |
| 16 | Y | BDL |
| 17 | Zr | 23.6±6.1 |
| 18 | Nb | BDL |
| 19 | Mo | BDL |
| 20 | Ru | BDL |
| 21 | Ag | BDL |
| 22 | Pb | 17.68±7.7 |


Table-5; G4 sample elements and
concentrations

| S.NO | Element | G4 |
|------|---------|-----|
| 1 | Cl | 403.5±19.5 |
| 2 | K | 4246±29.3 |
| 3 | Ca | 2754±28.1 |
| 4 | Ti | 13.67±11.3 |
| 5 | V | BDL |
| 6 | Cr | 16.93±2.5 |
| 7 | Mn | 33.68±4.1 |
| 8 | Fe | 5838±22.2 |
| 9 | Ni | 8.94±3.92 |
| 10 | Cu | BDL |
| 11 | Zn | 14.41±3.5 |
| 12 | Se | 0.9499±1.7 |
| 13 | Br | 13.47±6.2 |
| 14 | Rb | 34.71±6.2 |
| 15 | Sr | 27.65±5.2 |
| 16 | Y | 18.13±5.0 |
| 17 | Zr | 63.7±8.7 |
| 18 | Nb | 6.09±3.29 |
| 19 | Mo | 10.84±4.04 |
| 20 | Ru | BDL |
| 21 | Ag | BDL |
| 22 | Pb | 28.82±11.2 |




Table-6; G5 sample elements and concentrations

| S.NO | Element | G5 |
|------|---------|-----|
| 1 | Cl | 546.9±23.5 |
| 2 | K | 6699±40.2 |
| 3 | Ca | 4120±23.5 |
| 4 | Ti | 1590±13.8 |
| 5 | V | 37.78±6.0 |
| 6 | Cr | 17.71±3.1 |
| 7 | Mn | 47.43±5.3 |
| 8 | Fe | 7325±26.4 |
| 9 | Ni | 28.29±5.02 |
| 10 | Cu | BDL |
| 11 | Zn | 11.96. ±4.6 |
| 12 | Se | BDL |
| 13 | Br | 10.32. ±4.031 |
| 14 | Rb | 56.14 ±7.9 |
| 15 | Sr | 38.02±6.5 |
| 16 | Y | BDL |
| 17 | Zr | 11.44±7.0 |
| 18 | Nb | BDL |
| 19 | Mo | BDL |
| 20 | Ru | BDL |
| 21 | Ag | BDL |
| 22 | Pb | 41.58±14.48 |

Table-7; G6 sample elements and concentrations

| S.NO | Element | G6 |
|------|---------|-----|
| 1 | Cl | 383.7±16.7 |
| 2 | K | 5458±27.1 |
| 3 | Ca | 2544±26.2 |
| 4 | Ti | 1044±9.1 |
| 5 | V | 10.79±4.0 |
| 6 | Cr | 9.118±2.1 |
| 7 | Mn | 27.36±3.5 |
| 8 | Fe | 4905±17.7 |
| 9 | Ni | 10.11±3.1644 |
| 10 | Cu | BDL |
| 11 | Zn | 24.73±2.9 |
| 12 | Se | BDL |
| 13 | Br | 8.61±2.455 |
| 14 | Rb | 34.58±5.3 |
| 15 | Sr | 33.91±4.9 |
| 16 | Y | BDL |
| 17 | Zr | 12.16±5.0 |
| 18 | Nb | BDL |
| 19 | Mo | BDL |
| 20 | Ru | 9.977±3.59 |
| 21 | Ag | BDL |
| 22 | Pb | BDL |

Table-8; G7 sample elements and
concentrations

| S.NO | Element | G7 |
|---|---|---|
| 1 | Cl | 462.6±20.6 |
| 2 | K | 5393±33.4 |
| 3 | Ca | 3091±32.1 |
| 4 | Ti | 1510±12.4 |
| 5 | V | 11.55±12.4 |
| 6 | Cr | 14.13±2.6 |
| 7 | Mn | 27.03±4.5 |
| 8 | Fe | 6238±23.1 |
| 9 | Ni | 16.5±4.3692 |
| 10 | Cu | 8.717±3.47 |
| 11 | Zn | 4.29±2.0 |
| 12 | Se | BDL |
| 13 | Br | 9.08±3.503 |
| 14 | Rb | 52.46±7.1 |
| 15 | Sr | 35.53±6.1 |
| 16 | Y | BDL |
| 17 | Zr | 77.82±9.3 |
| 18 | Nb | BDL±6. ±6. |
| 19 | Mo | BDL |
| 20 | Ru | BDL |
| 21 | Ag | BDL |
| 22 | Pb | 24.49±12.5 |

The PIXE spectrum of the geological
samples G1 to G7 collected from the
interior of the Charnockite rock recorded
by Si (Li) detector. These concentrations
are presented with errors in Table-9
assuming Standard Deviation values (n) =
7 and BDL (Below Detection Limit).
Another attempt is made to analyze
the samples using an atomic absorption
photo spectrometer and same elements are
reported using the method of AAS and the
same standard also employed for a method
of AAS also. The data generated the AAS
method has been used to compare the
PIXE results for its evaluation purpose. In
this paragraph, each element is considered
in evaluating PIXE as per that element is
concentrated.
It is observed that the results
obtained by AAS are close which already
published data on Charnockites in various
journals. The possibility of increasing
accuracy of PIXE in analyzing samples of
complex matrix like Charnockite has been
discussed and suggestions are made.
So, the following tables presents the
elements which are close to elements
having moderate errors and elements
which are highly erroneous not detected.
Using this data, the Charnockite hill from
where the samples are collected have been
attempted to understand the chemical
nature followed by genetic implications.
The reasons behind the poor performance
of PIXE with respect to certain elements
have been tried to explain.
The following tables give overall
analysis of Charnockite samples by using
PIXE analysis and also AAS analysis.
Later the validity of AAS analysis verified
through wt% in the major elemental
components in the form of oxides in below
tables. Finally reviewed the Charnockite
composition, by using PIXE with
comparing results with AAS (tables 9-13).





282          Table- 9; Analytical results of all geological samples (PIXE)

| S.NO | Element | G1 | G2 | G3 | G4 | G5 | G6 | G7 |
|------|---------|-----|-----|-----|-----|-----|-----|-----|
| 1 | Cl | 394.1±16.5 | 399.6±17.1 | 379±16.5 | 403.5±19.5 | 546.9±23.5 | 383.7±16.7 | 462.6±20.6 |
| 2 | K | 4080±28.2 | 4187±25.1 | 4148±26.1 | 4246±29.3 | 6699±40.2 | 5458±27.1 | 5393±33.4 |
| 3 | Ca | 2229±25.9 | 2281±22.8 | 2637±25.6 | 2754±28.1 | 4120±23.5 | 2544±26.2 | 3091±32.1 |
| 4 | Ti | 1394±11.6 | 1271±9.3 | 1109±9.3 | 13.67±11.3 | 1590±13.8 | 1044±9.1 | 1510±12.4 |
| 5 | V | 17.92±4.8 | 23.69±4.1 | 7.85±4.0 | BDL | 37.78±6.0 | 10.79±4.0 | 11.55±12.4 |
| 6 | Cr | 16.63±2.3 | 38.53±2.1 | 15.16±2.1 | 16.93±2.5 | 17.71±3.1 | 9.118±2.1 | 14.13±2.6 |
| 7 | Mn | 18.62 ±3.9 | 34.71±3.7 | 34.31±3.7 | 33.68±4.1 | 47.43±5.3 | 27.36±3.5 | 27.03±4.5 |
| 8 | Fe | 5200±20.3 | 6575±21.0 | 5649±19.8 | 5838±22.2 | 7325±26.4 | 4905±17.7 | 6238±23.1 |
| 9 | Ni | 10.96±3.7 | 11.43±3.7 | 10.06±3.39 | 8.94±3.92 | 28.29±5.02 | 10.11±3.1644 | 16.5±4.3692 |
| 10 | Cu | BDL | BDL | 6.1±2.7 | BDL | BDL | BDL | 8.717±3.47 |
| 11 | Zn | 9.147±3.3 | 18.21±3.5 | 14.23±3.0 | 14.41±3.5 | 11.96. ±4.6 | 24.73±2.9 | 4.29±2.0 |
| 12 | Se | 6.3±65±2.5 | BDL | BDL | 0.9499±1.7 | BDL | BDL | BDL |
| 13 | Br | 4±2.2 | 12.17±2.93 | 11.82±2.85 | 13.47±6.2 | 10.32±4.031 | 8.61±2.455 | 9.08±3.503 |
| 14 | Rb | 48.87±6.0 | 42.08±6.1 | 62.73±6.0 | 34.71±6.2 | 56.14 ±7.9 | 34.58±5.3 | 52.46±7.1 |
| 15 | Sr | 38.5±5.4 | 28.6±4.6 | 44.61±4.9 | 27.65±5.2 | 38.02±6.5 | 33.91±4.9 | 35.53±6.1 |
| 16 | Y | BDL | 12.64±4.5 | BDL | 18.13±5.0 | BDL | BDL | BDL |
| 17 | Zr | 95.91±9.8 | 20.86±6.0 | 23.6±6.1 | 63.7±8.7 | 11.44±7.0 | 12.16±5.0 | 77.82±9.3 |
| 18 | Nb | 7.035±3.1 | 9.812±3.9 | BDL | 6.09±3.29 | BDL | BDL | BDL. |
| 19 | Mo | BDL | 24.34±6.3 | BDL | 10.84±4.04 | BDL | BDL | BDL |
| 20 | Ru | BDL | BDL | BDL | BDL | BDL | 9.977±3.59 | BDL |
| 21 | Ag | BDL | 12.36±9.0 | BDL | BDL | BDL | BDL | BDL |
| 22 | Pb | 32.93±15.6 | 38.35±17.3 | 17.68±7.7 | 28.82±11.2 | 41.58±14.48 | BDL | 24.49±12.5 |





Table-10; Analytical results of all geological samples (AAS).

| Element | G11 | G21 | G31 | G41 | G51 | G61 | G71 |
|---|---|---|---|---|---|---|---|
| Na | 23483 | 15561 | 22300 | 23718 | 26679 | 25505 | 25306 |
| Mg | 20568 | 16467 | 20026 | 20209 | 20501 | 24063 | 17546 |
| Al | 89081 | 90776 | 87760 | 96543 | 88132 | 89302 | 90105 |
| Si | 260631 | 274683 | 265398 | 269366 | 265905 | 262318 | 261506 |
| P | 1367 | 1402 | 916 | 1103 | 1231 | 1450 | 1582 |
| Cl | - | - | - | - | - | - | - |
| K | 24813 | 29466 | 27052 | 25739 | 9268 | 11374 | 13908 |
| Ca | 33709 | 28918 | 30776 | 29412 | 29418 | 36403 | 39305 |
| Ti | 7001 | 7403 | 7606 | 7842 | 6582 | 7409 | 7205 |
| V | 127 | 125 | 136 | 129 | 143 | 137 | 126 |
| Cr | 38 | 34 | 39 | 33 | 38 | 31 | 39 |
| Mn | 2023 | 2706 | 1855 | 1784 | 1548 | 2202 | 3011 |
| Fe | 87176 | 69983 | 88511 | 71342 | 90979 | 86904 | 87202 |
| Co | 8 | 10 | 7 | 9 | 7 | 11 | 7 |
| Ni | 38 | 34 | 42 | 45 | 39 | 37 | 42 |
| Cu | 8.8 | 12.8 | 11.9 | 9.2 | 6.6 | 8.6 | 11.1 |
| Zn | 11.3 | 12.3 | 9.3 | 15.4 | 10.6 | 10.9 | 9.3 |
| Se | 6.3 | 7 | 4 | 6 | 7.4 | 3.9 | 5 |
| Br | - | - | - | - | - | - | - |
| Rb | 60.8 | 70.1 | 74.6 | 89 | 82.9 | 77.5 | 71 |
| Sr | 144 | 132 | 143 | 126 | 119 | 128 | 148 |
| Y | 56 | 39.5 | 47.3 | 49.1 | 45.8 | 63.2 | 46.2 |
| Zr | 194.1 | 189.6 | 198.9 | 190.6 | 189 | 12.1 | 12 |
| Nb | 15.9 | 14.5 | 13.1 | 12.1 | 10.9 | 12.9 | 12.6 |
| Mo | 5.8 | 6 | 6 | 5.3 | 8 | 8 | 6 |
| Ru | 4.9 | 5.9 | 4.3 | 4.2 | 4 | 6 | 4.4 |
| Ag | 10 | 4.2 | 4.1 | 4.2 | 7 | 5 | 4 |
| Pb | 29.3 | 27.9 | 31.3 | 32.4 | 28 | 28 | 32.9 |
| Ba | 852 | 843 | 839 | 858 | 823 | 857 | 819 |







289          Table-11; Total Analytical results of all samples (AAS).

| Element (Wt.%) | G11 | G21 | G31 | G41 | G51 | G61 | G71 |
|---|---|---|---|---|---|---|---|
| Si | 26.063 | 27.68 | 26.539 | 26.936 | 26.590 | 26.23 | 26.15 |
| Al | 8.908 | 9.077 | 8.776 | 9.654 | 8.813 | 8.93 | 9.01 |
| Ti | 0.7 | 0.940 | 0.760 | 0.784 | 0.658 | 0.74 | 0.72 |
| Fe (total) | 8.717 | 6.99 | 8.851 | 7.134 | 9.097 | 7.88 | 8.72 |
| Mg | 2.056 | 1.646 | 2.002 | 2.020 | 2.050 | 2.406 | 1.754 |
| Mn | 0.202 | 0.270 | 0.185 | 0.178 | 0.154 | 0.22 | 0.301 |
| K | 2.481 | 2.946 | 2.7058 | 2.573 | 0.9268 | 1.137 | 1.39 |
| Ca | 3.37 | 2.891 | 3.027 | 2.941 | 2.949 | 3.648 | 3.93 |
| Na | 2.346 | 1.556 | 2.230 | 2.371 | 2.667 | 2.55 | 2.53 |
| P | 0.136 | 0.140 | 0.092 | 0.110 | 0.123 | 0.145 | 0.158 |



292          Table-12; Major Elements of all samples (AAS) Wt%.

| Element (wt.%) | G11 | G21 | G31 | G41 | G51 | G61 | G71 |
|---|---|---|---|---|---|---|---|
| $SiO_2$ | 55.81 | 58.82 | 56.83 | 57.68 | 56.94 | 56.17 | 56.01 |
| $Al_2O_3$ | 16.84 | 17.16 | 16.59 | 18.25 | 16.66 | 16.89 | 17.05 |
| $TiO_2$ | 1.17 | 1.57 | 1.27 | 1.31 | 1.10 | 1.24 | 1.21 |
| $Fe_2O_3$ | 1.65 | 1.39 | 1.72 | 1.094 | 1.289 | 0.81 | 0.82 |
| $FeO$ | 9.74 | 7.77 | 9.85 | 7.78 | 10.05 | 10.15 | 10.18 |
| $MgO$ | 3.41 | 2.73 | 3.321 | 3.35 | 3.42 | 3.51 | 2.91 |
| $MnO$ | 0.262 | 0.35 | 0.241 | 0.23 | 0.20 | 0.29 | 0.39 |
| $K_2O$ | 2.99 | 3.55 | 3.26 | 3.10 | 1.16 | 1.37 | 1.67 |
| $CaO$ | 4.72 | 4.05 | 4.31 | 4.12 | 5.42 | 5.11 | 5.31 |
| $Na_2O$ | 3.17 | 2.10 | 3.01 | 3.20 | 3.60 | 3.45 | 3.42 |
| $P_2O_5$ | 0.31 | 0.32 | 0.21 | 0.25 | 0.28 | 0.33 | 0.36 |
| Total | 100.078 | 99.81 | 100.423 | 100.84 | 100.58 | 100.57 | 100.27 |







297          Table-13; Analytical results of all geological samples (PIXE and AAS).

| S. No | Element | PIXE(Average values of seven samples, ppm) | AAS (Average values of seven samples, ppm) |
|---|---|---|---|
| 1 | Na | - | 23221.70 |
| 2 | Mg | - | 19911.42 |
| 3 | Al | - | 90242.71 |
| 4 | Si | - | 265686.71 |
| 5 | P | - | 1293 |
| 6 | Cl | 424.2±18.62 | - |
| 7 | K | 4887.28±29.91 | 20231.42 |
| 8 | Ca | 2808±26.31 | 32563 |
| 9 | Ti | 1133.09±10.97 | 7292.57 |
| 10 | V | 18.26±5.88 | 131.85 |
| 11 | Cr | 18.31±2.4 | 36 |
| 12 | Mn | 31.934±4.1 | 2161.28 |
| 13 | Fe | 5961.42±21.5 | 83156.71 |
| 14 | Co | - | 8.42 |
| 15 | Ni | 13.755±3.89 | 39.5 |
| 16 | Cu | 7.40±3.08 | 9.85 |
| 17 | Zn | 13.85±3.25 | 11.3 |
| 18 | Se | 3.65±2.1 | 5.65 |
| 19 | Br | 9.92±3.45 | - |
| 20 | Rb | 47.36±6.37 | 75.12 |
| 21 | Sr | 35.26±5.37 | 134.28 |
| 22 | Y | 15.385±4.75 | 49.585 |
| 23 | Zr | 43.64±7.41 | 140.9 |
| 24 | Nb | 7.6±3.43 | 13.14 |
| 25 | Mo | 17.59±5.17 | 6.442 |
| 26 | Ru | 9.97±3.59 | 4.20 |
| 27 | Ag | 12.36±9.0 | 5.5 |
| 28 | Pb | 30.64±13.13 | 29.97 |
| 29 | Ba | - | 841.57 |


299          Another attempt is made to analyze the samples using Atomic Absorption photo
spectrometer for PIXE fig; 1-7 evaluation or standardization Christopher et. al., (2016) in
high grade matrix composition and the same elements are reported and same standard also
employed for a method of AAS. The data generated the AAS method table-10 to 13 has been
used to compare the PIXE results Oti Wilberforce JO (2016) for its standardization purpose.
In this paragraph each element is considered in evaluating PIXE as per that element is
concentrated. It is observed that the results obtained by AAS are close which already
published data Rao and Babu  (1978) Sriramadas and Rao (1979) Charnockites in various
journals Saradhi (2000) Rajib Kar (2001) With respect to certain elements, the reasons behind
the improper performance of PIXE in case of matrix composition of geological materials
have been tried to explain Gerlad , et. al., (1993).



It is established that the recent advances in PIXE can successfully used in analyzing
samples with high accuracy, precision, low detection limits and high resolution from different
fields like geology of high grade metamorphic rocks. Low energy PIXE, high energy PIXE,
micro PIXE and external beam PIXE are recent developments which are used to determine
the various elements present in samples in the form of major, minor and trace amounts from
low to high atomic (Z) elements. During recent advances in PIXE, it is indicated that the
spatial resolution of peaks of various elements in samples of materials can be obtained in the
order of micro meter by developed micro PIXE and hence elimination of overlapping of
peaks in matrix effects are possible.
The detailed study table-1-13 of Charnockite samples from Visakhapatnam have been
established that there is certain accuracies pertaining to the concentration of certain elements.
Potassium is a major element in the Charnockite composition. The concentration of K in the
Charnockite composition by PIXE ranges from 4000 to 6000 ppm range in various locations.
But actually according to contents of Charnockite composition its value should be 20,000
ppm range. This is due to overlapped peaks of Ca K X-ray with K K X-ray in spectrum
obtained in this investigation. From this analysis PIXE is unable to detect major elements in
Charnockites of Visakhapatnam due to detector limits. Calcium is also a major composition
of Charnockite composition like potassium and in the analysis of Charnockites by using
PIXE, its value is 2500-3000 ppm range in all G1 series to G7 series samples. But in the
analysis of AAS its value is 30,000 ppm range. Therefore, PIXE once again fails to detect
exact value of ppm of major elements like K and Ca in Charnockite composition. The K and
Ca escape peaks in Si (Li) detector interfere with X-ray lines of Al, Pand S; K and Ca
summing peaks could interfere with X-ray lines of Fe, Ni and Cu. So in case of Charnockites
also high K and Ca contents, could be automatically resolved by GUPIX. The K and Ca
escape peaks in Si (Li) detector interfere with X-ray lines of Al, Si, P, Na and Mg; K and Ca
summing peaks could interfere with X-ray lines of Ti, V, Fe, Ni and Cu in Charnockite
composition.
The concentration of Ti in Charnockite by using PIXE method nearly 1200 -1400 ppm in
all types of samples. The error percentage of Ti is 3.6% only by standard reference. But
according to previous literature weight ratio in Charnockites it should be 7000 ppm and also
by present AAS analysis. From this PIXE analysis the error due to Ti-Kα and Fe-Kα have
closely 4.5 keV energy X-rays (4.509 and 4.647 keV) and also Ti-Ba have same X-ray
energies (4.469 and 4.509 keV). So PIXE not given proper value in case of major elements
like Ti due to above overlapping peaks. The value of Fe detection in Charnockite samples by
PIXE is around 6550 ppm but by the investigation through AAS its value is 86,000 ppm,
which is equal to wt% of Charnockite composition study. PIXE could not perform well in the
determination of Fe. Because the overlapped of Co-Kα and Fe- Kβ causes the inability of proper
detection and in addition to that Ti-Kα and Fe-Kα are same energy or nearly K X-ray energies,
(4.509 keV and 4.647 keV.) Some times it is difficult to resolve two peaks from neighboring
elements. Some corrections are needed to calculate the exact concentration value of Fe in
Charnockites by PIXE.



PIXE unable to detect the low Z elements present like Li, Be, F, Na, Mg Al, Si and P in
Charnockites due to detector limits. X-rays below or near the sodium cannot be seen because
they are absorbed in either the detector window atmosphere or though any filter used. A
possible disadvantage to running in this configuration is that low energy X-rays from lighter
elements attenuated in air. By increasing efficiency of detector or placing more suitable
detectors of by changing the incident proton beam energy or other technique like AAS used
in this investigation, these low atomic number (Z) elements are determined. Another reason
also is K and Ca escape peaks of in this investigation in Si (Li) detector interfere with X-ray
lines of Be, F, Na, Mg, Al, Si, and P Ana, et. al., (2011). The concentration of element V in
this PIXE analysis, 28-30 ppm and by AAS the value is 125ppm. From the above values
PIXE nearly gives results in case of minor elements and middle Z elements due to non-
overlapping of peaks and spectrum obtained by PIXE. The remaining 100 ppm due to Ti
overlapped peaks in the spectrums of Charnockite samples. So PIXE is more suitable in the
analysis of minor elements, middle atomic number (Z) elements. The error percentage of V is
around 15% through standard reference material.
PIXE results showed very good approximation in case of minor element .In case of Co
elemental analysis of Charnockite rocks by present PIXE investigation; the ppm value is not
detected. But Co trace presented in Charnockites according to previous literature and
composition of Charnockite by AAS. Non determination by PIXE of Co in Charnockites is
due to mainly two reasons; 1) Co- $K\alpha$ and Fe- $K\beta$ 2) Ni $K\alpha$ and Co $K\beta$ from the interferences
of above two peaks trace of cobalt not detected, but in case of AAS analysis Co also
presented in Charnockites and its value only is 8 to 10 ppm. In this case also Fe is the major
component so that the X-ray emitted from this element will dominate the energy spectrum, In
the PIXE analysis of Charnockites, the elemental concentration of Cu is very low and only
traces of one or two samples are obtained. But by using AAS analysis traces of Cu up to 12
ppm are present. The above reason is due to the interference of Cu peak with Zn peak in case
of Charnockites samples from Figure 3.1 to 3.7. It is Cu $K\alpha$ and Zn $K\beta$ interference matrix, i.
e, $K\alpha$ (Z+1) X-ray and the $K\beta$ (Z) X-ray and by correction small interference value, PIXE
once again proved in the detection case of trace elemental analysis.
By present PIXE analysis, the concentration value of Cr in Charnockite samples is 10-20
ppm and by AAS analysis, the value is also the same 40 ppm. Like V, in this case also
obtained the same result by PIXE analysis. Here no overlapping of peaks with V element in
the spectrums as shown in Figure 1 to 7. PIXE again given nearly good results in case of
minor elements as stated above in case minor elements, middle Z elements with small matrix
corrections. The concentration value of Mn obtained by PIXE analysis investigation is 20- 40
ppm and AAS analysis its value is 2000 ppm. In this case variation is due to the matrix of
non-resolution of peaks between Mn $K\beta$ and Fe $K\alpha$ overlapped. The error in this by using
standard reference material is 1%. So correction is required in case of Mn concentration
which is determined by PIXE analysis and also suitable to measure the concentration of Mn
major element with correction of matrix effects. Because in Charnockites Fe is the major
component. So the X-ray emitted from this element will dominate the energy spectrum as can
be seen from Figure; 1 to.7 and influence the Mn peak and concentration.


The concentration value of Ni by PIXE analysis is 10-20 ppm and ASS analysis is 40
ppm. In this element PIXE analysis given excellent result when compared with AAS and
remaining 20 ppm is due to the interference and non-resolution of peaks obtained by PIXE
with cobalt traces due to Ni Kα with Co Kβ. The error percentage in this case is 2% and
except 20 ppm correction compared to AAS and previous literature, PIXE given excellent
result in nickel minor elemental concentration of Charnockites, The concentration of Zn
element in samples of Charnockites by PIXE analysis, value is 10- 20 ppm range but the
results obtained by AAS its value is 10-15 in eastern ghats, Charnockites, Visakhapatnam.
The variation between the above two analysis, the error percentage of Zn though standard
reference is 16% and 5-10 ppm the variation between the above two analyses, which is very
low and good results obtained by PIXE analysis. So the variation of above value with AAS is
due the above Cu and Zn peaks overlapped in spectrums of Charnockites obtained by PIXE
analysis. In this case of Sr, the PIXE analysis 30-40 ppm range and AAS analysis 140 ppm
and difference 80 ppm, Sr Kα and Rb Kβ and also with Y Kα X-rays overlapped corrections
due to error value 3.5%. The PIXE had given the same results like above minor elements.
The concentration values of Rb obtained by PIXE and AAS are 40-50 ppm and 60 ppm. The
difference is nearly 20 ppm range and this is due to the overlapped peaks of Y Kα and Rb Kβ,
Sr Kα and R Kβ in the analysis of spectrum of the peaks of Charnockites. So by correct the
matrix20-30 ppm value of Rb and compared to AAS, PIXE has given good results in case of
Rb.
Using PIXE analysis, the concentration value of Y is 10-15 ppm which is low. From
Charnockite composition and present AAS analysis, its value is 60 ppm range and the
difference arises due to Y Kα and Rb Kβ overlapped of peaks and matrix effects. From the
above two elementsif the concentration value increases, PIXE performance value decreases.
By PIXE analysis the concentration value is 90 ppm and by AAS analysis its value is 190
ppm. From these analyses PIXE analysis is very poor in the detection of Zr and resolution of
peaks with Nb. The poor results obtained in PIXE analysis of element Zr in the Charnockites
as concentration value of element in Charnockite composition increases, poor results also
increase in PIXE analysis. It is same as above two cases. The error value increases with as
concentration value increases.
The concentration value of Nb in Charnockites is 5-11 ppm in PIXE and in case of AAS
its value is 15 ppm range, the concentration variation is only 10 ppm and it is due to small
matrix effect of Zr and Mo. Of course, PIXE analysis is good in case of trace elements like
Nb except small corrections due to resolutions of peaks and matrix effects. The element Pb is
an important element in rock formation, which is determined by PIXE and its value 20-30
ppm range and in the investigation of Pb, PIXE given good results. It gives valuable
information in rock crystallization process. In PIXE analysis, the concentration value
identified is only 1-2 series samples of Charnockites. So in PIXE analysis traces of Mo not
exactly found due to corrections of interferences with Nb but its value by AAS is also given
very low traces due to matrix effect of Nb.



The traces of Ru found by using PIXE like Ag in Charnockites. AAS analysis is also given
only traces of Ru in one or two sample series of Charnockites.It is a precious metal
previously not detected in Charnockite, in this PIXE also not trace out exactly because in ppb
levels and most of above series samples are at BDL. So PIXE is suitable to measure the
concentration of precious metal like Ag. The traces of Ag investigated also in the AAS
analysis. It is an important element in Charnockites in Visakhapatnam and these are halogen
rich minerals. PIXE is a very suitable technique in the determination of gaseous phase
elements like Cl. Its value is 400 ppm which is the similar result obtained by previous
analysis. But AAS is not able to detect the Cl element directly.Br is also a trace element in
halogen rich minerals of Charnockites. PIXE is very suitable in the detection of Br and Cl but
not F present in the composition due X- ray energy limitation. The gaseous elements give
information about rock formation. The traces of Br identified by the PIXE are in the range 5-
13 ppmand Br is not detected by AAS. PIXE is very much suitable technique in case of
middle Z elements like Zn, Rb, Cr, gaseous elements Cl and Br, in the detection of precious
metals Nb, Mo and Ru and other elements with corrections.
The presented PIXE technique is known for its sensitivity, accuracy, precision simplicity
and fast of thick target preparation and to perform multi elemental analysis of a large number
of complex geological materials like Precambrian, proto crustal rocks compared with the
previous study EPMA. By the identification of elements from upper mantle, crust and the
trace elemental data show that rock samples analyzed have a composition different from that
of the host Charnockite. These include the large ion lithophile elements (Rb, Sr) high field
strength elements (Ti, Zr, Nb) precious elements Ag, Cu and various elements of importance
in geo thermo barometry. However for REE elements, the LOD are markedly poorer. This is
particularly the case for the REE s Hajivaliei, et. al., (2000):
The elements K, Ca, Rb and Sr present in high concentration, low value of Ni in the
samples supports calcium alkaline phase. The element Fe has the highest concentration, Cr,
Zn and Zr are detected in the samples strongly indicates the ultramafic percentage from upper
mantle. In this aspect these rocks are entirely different from the host Charnockite and these
were evolved from upper mantle to crust during Precambrian period. The light and medium
heavy elements therefore determined by their K X-ray as the heavy ones by L X-rays which
are shown in figures. PIXE has certain drawbacks in detection of light elements Be, F, Na,
Mg, Al, Si, and P and the REE. For light elements, the problem is mainly due to absorption of
the low energy X-ray Durocher, et, al., (1988) by detector Si (Li) windows and the tails of the
heavier major elements peaks.
PIXE, especially with micro beam, has proven to be a versatile and powerful analytical
tool in many areas of geology. PIXE within a short span of time has demonstrated its
versatility and usefulness by the diversity of applications to which it has been applied. The
detection limits of the PIXE measurements were calculated assuming that the minimum
detectable peak is three times the square root of the background at FWHM. The detection
limits in the PIXE measurements 0–1ppm by weight with the exception (from all the tables
and values presented). We are aware of no attempts having been made to apply PIXE analysis



to the petrogenesis of intermediate to silicic igneous rocks. Our results show that such studies
are feasible.
The ability to analyze diverse phases using a single standard and uniform operating
conditions largely obviates the need to perform laborious mineral separations, formerly
necessary for studies of the trace-element inventory of such rocks. Among the most exciting
potential applications of PIXE analysis to silicate minerals and element-partitioning
measurements relevant to mineral melt, mineral-mineral, or melt-melt systems useful in
understanding melting and crystallization processes, trace element zoning studies in igneous
or metamorphic volcanic minerals to assess the crystallization behavior of magmas or to
assess tectonic and metamorphic histories and diffusion-profile measurements in minerals,
melts.
In the coming years PIXE should find itself being applied more and more to a variety of
samples, when a rapid non-destructive and multi-elemental method is looked for and it should
take its rightful place as an equal or better technique among the numerous other standard
analytical techniques available. We also demonstrated that nuclear microscopy, due to its
high spatial resolution and the low detection limits proved to be a powerful tool for the
characterization of minerals with complex chemistry and it serves as an ideal complementary
technique of optical mineralogy, SEM and EPMA. During to its longer probe depth (ten of
micro meter) the PIXE can be performed for a large range of elements ($K < Z < Pb$) in the
periodic table. So PIXE useful fast, precise, accurate and sensitive (limit in ppm). Both
matrix and trace elements PIXE gives higher signal to background ratio as compared to
EPMA and spectrum 100 times better sensitive and has higher resolution like previously
stated.
Compared to electron based X-ray analytical techniques such as energy dispersive
spectroscopy(EDS) and EPMA, PIXE offers better peak to noise ratios and consequently
much higher trace element sensitivities as seen in spectrum Figure 1 to 7. PIXE is capable of
multi-elemental analysis and a large number of elements (Twenty two) may be seen
simultaneously. PIXE, being a non-destructive technique and because the original shape and
size of the sample is not destroyed, makes it a unique facility for a number of applications.
PIXE and XRF are both methods based on X-ray emission and have several features in
common. From sensitivity point of view, PIXE has certain superiority, moreover the
bremsstrahlung produced in PIXE is a secondary effect where as in case of Electron
Microprobes and XRF it is primary contributor and the principal source of proton back
ground against which the character X-rays of elemental constituents must be distinguished
and hence is also the principal determinant of detection limits. The low bremsstrahlung in
PIXE enables parts per million sensitivities, superior to its sister techniques in geological
materials are obtained (From previous literature and previous EPMA and XRF).
The results obtained indicate that it is competitive with other more classical analytical
methods, and that it may be, in addition, a very useful complementary technique when
combined with other ion beam methods like PIGE, EPMA, XRF and NAA is, in general, a
very sensitive method but has limited use for light elements. Also, Particle-Induced X-ray
Emission (PIXE) fails in the situation where the species of interest has a low atomic number
because the low K X-ray fluorescence yields are strongly attenuated by the absorption edge
of higher atomic number elements present in the sample. Prompt γ-ray analysis (PIGE) offers
an alternative in measuring light elements which are not detected by PIXE and has the
advantage that γ-rays from the different light elements can be easily distinguished by their
energies. The sensitivity of PIGE analysis can be improved by coincidences measurements.
From this it results that by coupling PIGE, PIXE and NAA methods, a very good overall
picture of the elemental composition of a complex target such as steel may be obtained. PIXE
and NAA are complements of PIGE when the determination of medium and heavy elements
with high sensitivity is necessary in geological materials. Neutron activation is a very
efficient tool of high sensitivity, but it is expensive and seldom used; also this technique is
not convenient for detection of some elements of great interest in geochemistry such as Al, Si
or Pb. But detection of Pb possible in this PIXE, off course light elements like Al, Si not
detected but light elements are possible by using PIGE.
The method is a very sensitive one having a minimum detectable concentration of about
0.1–1 ppm under optimum conditions. In the coming years PIXE should find itself being
applied more and more to a variety of samples, when a rapid non-destructive and multi-
elemental method is looked for and it should take its rightful place as an equal or better
technique among the numerous other standard analytical techniques available. PIXE analyses
of silicate samples are of excellent quality for a broad range of elements at abundance levels
of one to hundreds of parts per million, depending on counting times. PIXE with Si (Li)
detector not suitable to detect REE because in the determination of REE in geological
samples is a very important subject but hard to tackle. By PIXE the L X-rays energies from 4-
9 keV strongly overlap with K X-rays energies of light elements (20 to 30 Z).
The quantification of REE by PIXE becomes very difficult and inaccurate as results of the
needed complex spectrum deconvolution. In fact in the context of REE L X-ray using
standard PIXE set by compared results obtained with PIXE, XRF and NAA for geological
samples and concluded that with the PIXE technique the limits are 30 ppm for REE, higher
than with other techniques as NAA. The use K X-ray of REE elements could be a solution
but at that energies Si (Li) detection can no longer is used. To overcome this problem the use
of large Ge detection to detect the L X-ray of REE was attempted but due to large
dimensions, a large Compton back ground is present which degrades the detection limits in
the high X-ray energy region. Small Ge detection represent in improvement relative to this
problem. But their overall size is still significative to this problem. But their overall crystals
reduced solid angle Cd Te detects have average atomic numbers of 50 allow the construction
of small detection.
PIXE roughly analyses the element whose atomic numbers, the light and medium-heavy
identified by their K X-ray and the heavy elements by L X-ray due to the effective detection
of the K X-ray which can be obtained in the range 20<Z<50, L X-ray for Z>50 and20 <Z<50



for M X-ray. So due to above reason Na, Al, Si, and Mg not detected due to low Z elements
(Z<17) in this investigation.
PIXE fails in the situation where the species of interest has a low atomic number because
the low K X-ray fluorescence yields are strongly attenuated by the absorption edge of higher
atomic number elements in the present investigation. From the above results, PIXE at IOP,
Bhubaneswar, India, Charnockites elemental analysis lower atomic number(Z) in all
spectrums is Cl (Z=17) and higher atomic number (Z) in all spectrums is Ag (Z=47) for K X-
ray. For L X--rays only Pb is present in charnockites (Z greater than 50).The comparison with
roughly PIXE (20<Z<50 for K X-ray and Z>50 for L X-ray), good results obtained by PIXE
in the elemental analysis of Chatnockites of airport hill Visakhapatnam.
The element Sc is a trace which is detected previously by nuclear techniques other than
PIXE. But in this investigation above element is not detected due to the X-ray Kα=4.093 keV
is equal to the Kβ=4.013 keV of Ca and hence resolution of Si (Li) detector at 160eV at
5.9keV. Simillarly Kβ=4.464 keV of Sc is equal to Kα=4.512 keV of Ti. The element Y is
also trace is not detected properly by present PIXE because Y Kα=14.958 is equal to Rb
Kβ=14.961 keV and Y Kβ=16.739 keV is equal to Nb Kα=16.615 keV.
Again from this analysis, PIXE once again proved with good accuracy with ±11% in the
detection of elements present in samples, better resolution in case of all elemental peaks with
detection limits <1 ppm with precious trace elemental detection through sensitivity. Non
detection of Th and U by PIXE in this investigation depend detector window. It can be
performed for a large range of elements (K<Z<Pb). The lower limit started from Z= 17(Cl) is
due to experimental detector limits because PIXE analysed the elements present in samples
from Z=17(Cl) to Z=Pb (82) (from previous literature and present investigation also). So
eventhough some elements are present in eastern ghat Charnockite from previous analysis
like XRF and EPMA, here not detected because of detector efficiency. Ba was detected
previously in the major element category of Charnockites like XRF analysis but it is not
possible due to following explanation. In case of PIXE, the light and medium-heavy elements
are identified by their K X-rays and the heavy elements by L X-rays. Due to the effective
detection of the K X-ray which can be obtained in the range 20<Z<50 and of the L X-ray for
Z>50. Since the value of atomic number (Z) of Ba is 56, this element is not suitable to detect
by PIXE. Similarly, the middle Z elements Sc, Ce, Sn, W, Ge, Ga and also Au not detected
due to overlapped.

### 3.1. Rare Earth Elements Interpretation

Non detection of REE elements is due non suitability of Si (Li) detector to the REE
elements in PIXE. These samples belong to a very important geological phase and further
work on petrography and REE of the rock is indeed to firmly establish its exact parentage.
PIXE has weaknesses in detection of media Z elements (Ti, V, Mn, Cr, Fe, Ni, Co) and the
REE. For light elements, the problem is mainly due to absorption of the low energy X-ray by
detector Si (Li) windows and the tails of the heavier major elements peaks. One approach,





such as that will be tried to use ultra thin window for X-ray detectors. This enables to extend
the lower limit of detectable atomic number but measures must be taken to protect the
detector from scattered proton beams, for example by using magnetic beam deflector.
Poor REE detection is due to the fact that the L-X-ray are either absorbed heavily or
interfered with by the Kα line of Fe which is present in most geological rock samples with
low energy proton. The K and L X-ray production cross section and intensity ratios of rare-
earth elements for proton impact in the energy range 20–25 MeV are useful for detection of
trace elements in rare earth region due to their high K X-ray ionization cross-section
Hajivaliei, et. al., (2000). The use of their K X-ray is not practical because of the low yield
and the low efficiency of the Si (Li) detector for X-ray>30 keV. Improvements in the future
would address these problems and include the use of Ge detector or WDS detector in
principle. PIXE, being a non-destructive technique and because the original shape and size of
the sample is not destroyed, makes it a unique facility for a number of applications for the
determination of REE of samples needed.
**3.2. Other Traces Elemental Interpretation**
The trace of element bromine is not all detected previously in Charnockite hill of
Visakhapatnam by previous methods like EPMA and XRF. This investigation by PIXE
clearly proved that this technique could detect not only regular elemental phase Cl in halogen
rich minerals of Charnockites, and also the trace of Br. This aspect is highly interesting and
potential geological applicability in similar investigations. From these fluids containing rich
in Cl, F and also trace of Br are recognized to have contributed significantly to the evolution
of proto crust. In this PIXE technique element, F is not detected due to PIXE has weaknesses
in detection of light elements (F, Na, Mg,, Al, Si, and P).
Here the silver identified in some few samples, it is an interesting investigation. Previous
methods not identified this type of precious metal in analysis and no method from
Charnockite analysis identified the precious metal Ag. So PIXE is wonderful technique used
to identify the traces of Ag in Eastern Ghat Charnockites particularly in high metamorphic
rocks. PIXE is a sensitive technique in the analysis elements even though if they are present 1
micro gram/gram through scan the sample even small quantities are present in few samples.
Nb is also present in few samples in the order of below 10 ppm. From this analysis PIXE has
higher sensitivity with accuracy without destroy the sample and is very important in case of
precious metals like Nb. Previous analytical techniques like XRF and EPMA except PXE,
trace of Cu not detected in Charnockites of Visakhapatnam. The PIXE once again proved in
case of traces present even though if they are present in one or two samples in ppm levels.
The importance of PIXE technique is used to find out the new trace elements if they are
present even in small quantities at any corner of sample even in micro gram. Previously in
Eastern Ghats analysis in any method Se and Mo not identified but in this investigation they
are in clearly appreciable quantities in few samples. This kind of experimental research work
may be compared with the previous work published in elsewhere, accuracy evaluation





absolute calibration in thick target PIXE. Data have been compared with the standard used
mainly USGS. Similarly data are compared with elemental concentration obtained for the
samples studied using another standard technique AAS. In the above paragraphs the PIXE
has been evaluated based on the data pertaining to various elements determined. The
evaluation has been made and comparing the accuracies of elemental concentration with
respect to Charnockite sample. The traces of Th, U also present in charnockite composition,
in the PIXE analysis not detected due to X-ray energy limits. Because of wide range of
elements from low atomic number Z to high atomic number present in matrix composition of
geological materials and characteristic X-rays of any series of elements present are at similar
to characteristic X-rays of thorium and uranium in composition.

## 4. Conclusions

In support of our observation also the theoretical back ground behind on performance of
PIXE against induced elements has been discussed in detail. An analysis is made to
understand the results behind the poor performance of PIXE with respect to certain elements.
The possible reasons behind this have been brought out.
The general observation of PIXE methodology indicate that PIXE has been operated at
one 3MeV proton energy with such condition the determination of different elements from
low Z to high Z is not possible and perhaps this is the most important reasons behind the poor
performance of PIXE with respect to the certain elements. It is suggested that with an
investigation of PIXE by analysing pure samples of problematic elements. It means them
100% pure elemental powders needed to be analysed with different concentrations at
different levels using a non interfering matrix so that the optimum conditions for that element
can be obtained. Further work is needed to understand the problems pertaining to
combinations of elements. Such experiment is needed to design to determine different
elements in a sample in combination to an optimum condition of analysis. In this way the
conditions of PIXE can be standardized for low Z to high Z elements under different
combinations. This kind of experimental research is highly necessary to fine tune the
performance of PIXE especially when dealing with materials of complexity like high grade
metamorphic such as Charnockite.
The present study could establish this aspect for the first time which will help in future
for effective analysis of complex samples using PIXE. This experiment should contribute the
conditions of PIXE operation for materials of different complexity and matrix.
The presented PIXE technique is known for its sensitivity, accuracy, precision simplicity
of thick target preparation and to perform multi elemental analysis of a large number of
complex geological materials, like Precambrian, proto crustal rocks. This investigation
clearly proved that this technique could detect not only these elemental phases but also the
concentrations of gaseous elements like Br, Cl. And K, Ca, Rb, Sr present in high
concentration in the samples supports calcium alkaline phase and also Fe has the highest
concentration, Cr, Zn, Zr are detected in the samples strongly indicates the ultramafic





percentage from upper mantle. This aspect is highly interesting and has partial geological
applicability in similar investigations

**Acknowledgment**

I deem it a privilege to express my deep sense of gratitude and heartiest thanks to my
research director. I am thankful to Prof A. Durga Prasad Rao, Head of the Department,
Department of Nuclear Physics, Andhra University, Visakhapatnam and teaching staff. The
deep sense of gratitude and heartiest thanks to my Head of the Department, Engineering
Physics, Andhra University, Visakhapatnam. I thank Dr. D. P. Mahapatro, Director, Institute
of Physics, and Bhubaneswar, providing accelerator facility to carry out the research work
and for all amenities provided during our stay at the institute of physics.

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

of major, minor element PIXE analysis of rocks and mineral. Nuclear Instruments and
Methods in Physics Research Section B: Beam Interactions with Materials and Atoms. 366,
pp. 40-50.
3) Durocher, J. J. G., Halden, N. M., Hawthorne, F. C and Mckee, J. S. C., (1988). PIXE and
micro PIXE analysis of minerals at Ep=40 MeV.Nucl. Instr. Meth. Phys. Res. B, Vol. 30, pp.
695 470-473.

4) French, J. E., Heaaman, N. M., and Srivastava R. K., (2008). Ma southern bastar
Cuddapah mafic igneous events, India.A newly recognised large igneous province,
Precambrian research,Vol. 160, pp 308-322.
5) Gerlad, K Czamanske., Thomas, W Sisson., John. L Campbell., William, J Teesdale.,
(1993). Micro-PIXE analysis of silicate reference standards. American Mineralogist. 78, pp.
701 893-903.

6) Hajivaliei, M., Puri, Sanjiv., Garg, M. L., Mehta, D., Kumar, A., Chamoli, S. K., Avasthi,
D K., Mandal, A., Nandi, T K., Singh, K P., Singh, Nirmal., and Govil, I. M., (2000). K and
L x-ray production cross sections and intensity ratios of rare earth elements for proton impact
in the energy range 20-25 MeV Nucl. Instr. Meth. Phys. Res. B, Vol. 160, pp. 203.
7) Kullerud, G., Steffen, R. M., (1979)."Proton induced x-ray emission (PIXE): "A new tool
in geochemistry",Chemical Geology, Vol. 25, pp. 245-256.



8) Luciana, V. Gatti., Antonio, A Mozeto., and Paulo, Artaxo., (1999). "Trace elements in
lake sediments measured by the PIXE Technique  Nucl Instr. and methods in physics
research  B, Vol. 150, pp. 298-305.
9) Malmqvist, K. G., Bage, H., Carlesson, L. E., Kristiansson, K., and Malmqvist, L.,
(1987)."New methods for mineral prospecting using PIXE and complementary techniques."
Nucl Instr. and  methods in physics research B, Vol. 22, pp. 386-393.
10) Oti Wilberforce, J. O., (2016). Review of Principles and Application of AAS, PIXE and
XRF and Their Usefulness in Environmental Analysis of Heavy Metals. IOSR Journal of
Applied Chemistry. 9, Issue-6, PP. 15-17.
11) Rajib, Kar., (2001). Patchy Charnockites from Jenapore, Eastern Ghats granulite belt,
India: Structural and petrochemical evidences attesting to their relict nature India. Proc.
Indian Acad. Sci. (Earth Planet. Sci.) 110, No. 4, pp. 337–350.
12) Rao, A. T., and Babu, V. R. R. M., (1978). "Allanite in charnockites rock airport hill near
visakapatnam, andhrapradesh, American mineralogist, Vol. 63, pp. 330-331,
13) Radha Krishna, B. P.,(2008).  Forewood-precambrian mafic magmatism in the   Indian
shield., Jour, Geol. Soc.India.,V. 72, pp. 5.
14) Srivastava, R. K., and Ahmad, T., (2008). Precambrian mafic magmatism in the Indian
shield, an introduction: Jour. Geol. Soc. India., V. 72, pp. 9-13.
15) Saradhi, M., Arima, M., Rao, A. T., Yoshida, M., (2000). Whole rock geochemistry of
massive and porphyritic Charnockites from the central part of the Eastern Ghats Belt, India
Journal of Geosciences, Osaka City University. 43, Art. 10, pp. I77-191.
16) Sriramadas, A., and Rao, A. T., (1979) Charnockites of Visakhapatnam, Andhra Pradesh.
Journal of Geological Society of India. 20, pp. 512-517.
17) Sie, S. H., Ryan, C. G., Consens, D. R., and Griffin, W. L., (1989). Nucl Instr. and
methods in physics research B. Vol. 40/41  pp. 690-697.
18) Tangi, S. M., Orlic, I., WU, X. K., (1998). "Analysis of Singapore marine sediments by
PIXE: Nucl. Instr. and methods in physics research B, Vol. 136-138, pp. 1013-1017.

