# Peer review of "Review on Mineral Characterization of Precambrian Charnockites-using PIXE Technique"

_Geoscientific Instrumentation, Methods and Data Systems, 2020_

## Referee Comment (RC1) · Anonymous Referee #1 · 2 Sep 2020

General comments:

This paper is interesting but requires significant editing, as it is not written in sound English. If possible please give a small table which will make it easy for the reader to see certain differences between techniques (as mentioned in the abstract).

List of some specific comments that would help to improve the manuscript:

- Line 10. Sentence Âńhas been applied to an analytical toolÂż should be revised.

- Line 11. What is REE named after? Please, consider writing its full name the first time you mention it in the manuscript. Acronyms should be extended with their full names

the first time they are cited in the body of the text.

- Line 18. Sentence Âńthere presentÂż should be revised. Maybe to Âżthey are present Âż

- Line 19. Sentence Âńwhich is operation at..Âż should be revised.

- Line 70. What is GUPIX named after? Please, consider writing its full name the first time you mention it in the manuscript. Also please consider citing the software if possible

- Line 78. Consider using Âńin the pastÂż or something similar instead of ÂńearlierÂż .

- Line 113. Typo: Âń._ÎďÂż instead of Âń.ÎďÂż

- Line 115. Sentence Âńis made to fall on to the sampleÂż should be revised.

- Line 115. Âńcharge collectedÂż should be revised. Do you mean Âń The charge is collectedÂż? In that case can you expand the sentence so as it is clear what exactly is the meaning of ÂńchargeÂż ?

- Line 115. What is the beam current?

- Line 116-117. Sentence should be revised. I find it hard to understand the meaning please check syntax.

- Line 137-139: Sentence should be revised.

- Figures 1-5: Figures have different sizes.

- Line 310: Sentence should be revised maybe try: Âńcan be successfully usedÂż

- Line 449: Phrase Âńand fastÂż should be revised. Fast compared to what?

---

## Short Comment (SC1) · 2 Sep 2020

Sir, Thank you very much for your valuable suggestions, the suggestions will be very useful for this paper. According to your comments we will correct some words/sentences of the paper. This paper is interesting but requires significant editing, as it is not written in sound English. If possible please give a small table which will make it easy for the reader to see certain differences between techniques (as mentioned in the abstract).

Corrections;-

[Figure]

Nuclear techniques in the elemental analysis are very familiar, if it is necessary in the inclusion of this paper, definitely I will include according to your valuable suggestion. List of some specific comments that would help to improve the manuscript: - -Line 10. Sentence Ânhas been applied to an analytical tool ' z should be revised. Answer;- has been applied to long range -Line 11. What is REE named after? Please, consider writing its full name the first time you mention it in the manuscript. Acronyms should be extended with their full names C1 GID Interactive comment Printer-friendly version Discussion paper the first time they are cited in the body of the text Answer;- Rare Earth Elements - Line 18. Sentence Ânthere present ' z should be revised. Maybe to  ˏ zthey are pre- ËŹ sentÂz Answer;-There are present - Line 19. Sentence Ânwhich is operation at.. ' z should be revised. ËŹ Answer;-operating - Line 70. What is GUPIX named after? Please, consider writing its full name the first time you mention it in the manuscript. Also please consider citing the software if possible Answer;-Guelph PIXE Software Package - Line 78. Consider using Ânin the past ' z or something similar instead of  ˏ nearlier ' zËŹ . Answer;- in the fast - Line 113. Typo: Ân._Îd' ' z instead of  ˏ n.Îd' ' zËŹ Answer;- Sir, with in bracket only - Line 115. Sentence Ânis made to fall on to the sample ' z should be revised. Answer;- onto ËŹ - Line 115. Âncharge collected ' z should be revised. Do you mean  ˏ n The charge is ' collectedÂz? In that case can you expand the sentence so as it is clear what exactly is ËŹ the meaning of Âncharge ' z ?ËŹ Answer;- charge is collected - Line 115. What is the beam current? Since it is a Proton beam and positive charge flow means current - Line 116-117. Sentence should be revised. I find it hard to understand the meaning please check syntax. Answer;-not exited "excited" in the sentence - Line 137-139: Sentence should be revised. Answer;- not different elements thus "different elements, thus" only coma mistake - Figures 1-5: Figures have different sizes. Answer;- I will arrange the figure in same size. - Line 310: Sentence should be revised maybe try: Âncan be successfully used ' zËŹ Answer;- "can be successfully" - Line 449: Phrase Ânand fast ' z should be revised. Fast compared to what? Answer;- fast means "speed"

Thanking You Sir,

Yours Obediently,

Author#1, gi-15-2020

---

## Referee Comment (RC2) · Anonymous Referee #2 · 5 Nov 2020

In this paper is studied the possibilities of Particle Induced X-Ray Emission technique (PIXE) to characterize Charnockites from Visakhapatnam samples (India). It shows an exhaustive analysis comparing the results from PIXE with Atomic Absorption Spectrometry (AAS). The analysis includes elements presented in the rocks in low concentrations down to ppb (parts per billion). The study is interesting and can open a wide range of future studies.

Despite this, authors need to fix some errors and explain carefully some details:

* The experimental details are properly explained and the data analysis is made with some software by a nonlinear least square algorithm. But no reference to any fitting

parameter such as $R^2$ is mentioned in the text. This should be included to have an idea of how good the data fits the curves.

\* Figures from 1 to 7 have been stretched horizontally and text in them is hardly read. Authors should consider to clarify this.

\* In lines 317 and 318 authors explain that by using modern advances in PIXE it is possible to eliminate the overlapping of peaks. But later, in the same page, they explain that the underestimation of concentration of certain elements can be explained by the overlapping of such peaks. Can this overlapping be eliminated or not? Authors must clarify this.

\* I encourage the authors to include the references with name and year between parentheses ("Precipitation increase was observed (Smith, 2009)..." ), unless they were part of the sentence ("As we can see in the work of Smith (2009) the precipitation has increased"). Please read author indications in the web (https://www.geoscientific-instrumentation-methods-and-data-systems.net/submission.htmlreferences)

\* Some typos need correction:

- Line 29 at the end: Phanerozoic

- Line 30 at the end: Cenozoic (or Caenozoic)

- Line 32 at the end: Phanerozoic

- sentence in lines 36 to 38 has no meaning.

- Also the sentence of lines 38 and 39 need to be rewritten.

- Sentence at lines 42 and 43 is not clear, it may need an "AND" between India and "IS" that according to the sentence should be "ARE".

- Line 45 at the end: need an "IS" somewhere.

- Line 50: "...The rocks are FROM Precambrian age..."

- Line 89: consider use precision rather than precession.

- Line 124, at the end: consider using annotated rather than noted (famous)

- Sentence in lines 139, 140 and 141: This sentence cannot be understood.

- Line 181: consider editing the sentence to chage the use of which ("... obtained by AAS are close to already published data ...)

- Table 12: Oxygen must be capilalized. ($SiO_2$)

- Line 301 at the end needs a verb.

- Line 310 at the end also needs a verb.

- Line 320: It cannot be used the form "there is" paired with "accuracies".

- Line 342: The sentence at the end needs some verb.

- Line 416: elements if (space)

- Line 643: background (eliminate space)

- Sentence beginning at the end of line 651: the use of "them" is not clear, and verb tenses must be related (past or present)

- Sentence beginning at line 655 cannot be understood.

- The end of sentence in lines 659 and 660 is not clear: high grade of metamorphic rocks? High grade of metamorphism?

- Sentence in lines 662 and 663 cannot be understood.

\* Some parts of the text need to be rewritten because they are not clear. Specially the Abstract and the Conclusions are hard to be understood due to the lack of verbs. I encourage the authors to make an exhaustive revision of all the sections to avoid so much grammar errors.

---

## Author Comment (AC1) · 7 Nov 2020

Sir,

Thank you very much for both reviewers about their valuable comments on PIXE analysis - Matrix Composition. First of all, we appreciate reviewer suggestions and contribute to improving the manuscript. According to reviewer comments, we already edited and corrected (sent through SC)significantly at some grammatical errors in the manuscript after Reviewer #1 comment and should be included in the manuscript of the final version with following Reviewer # 2 comments also. The remarks were implemented into the new version of the paper. We agree with the two reviewers for significant corrections at grammatical errors.

Anonymous Referee #2

 In this paper is studied the possibilities of Particle Induced X-Ray Emission technique (PIXE) to characterize Charnockites from Visakhapatnam samples (India). It shows an exhaustive analysis comparing the results from PIXE with Atomic Absorption Spectrometry (AAS). The analysis includes elements presented in the rocks in low concentrations down to ppb (parts per billion).The study is interesting and can open a wide range of future studies.

Despite this, authors need to fix some errors and explain carefully some details:

Reply to reviewer #2 comments

Despite this, authors need to fix some errors and explain carefully some details: * Q.1.The experimental details are properly explained and the data analysis is made with some software by a nonlinear least square algorithm. But no reference to any fitting. This should be included to have an idea of how good the data fits the curves? A; Sir, We included reference Q.2. Figures from 1 to 7 have been stretched horizontally and text in them is hardly read. Authors should consider clarifying this? A; No, specific issue, for page setting purpose only Q.3. In lines 317 and 318 authors explain that by using modern advances in PIXE it is possible to eliminate the overlapping of peaks. But later, in the same page, they explain that the underestimation of concentration of certain elements can be explained by the overlapping of such peaks. Can this overlapping be eliminated or not? Authors must clarify this? A; In case of complex matrix, we can't achieve complete results of total composition. Otherwise if it is simple or non matrix it is ok (at media Z) except at low Z and higher Z. element. Q.4 I encourage the authors to include the references with name and year between parentheses ("Precipitation increase was observed (Smith, 2009)..." ), unless they were part of the sentence ("As we can see in the work of Smith (2009) the precipitation has increased"). Please read author indications in the web (https://www.geoscientificinstrumentation-methods-anddata-systems.net/submission.htmlreferences) * A; Yes, Sir. Q; 5. Some typos need correction: • Line 29 at the end: Phanerozoic • Line 30 at the end: Cenozoic (or Caenozoic) • Line 32 at the end: Phanerozoic • ? A; Spellings Corrected Q; 6. Sentence in lines 36 to 38 has no meaning. • Also the sentence of lines 38 and 39 need to be rewritten. • Sentence at lines 42 and 43 is not clear? A; Yes, Previously corrected all grammar errors, according to respected Reviewer #1 suggestions. Q; 7. it may need an "AND" between India and "IS" that according to the sentence should be "ARE"? Q; 7. • Line 45 at the end: need an "IS" somewhere? A; and Corrected are, Corrected Q; 8. Line 50: "...The rocks are FROM Precambrian age. . ."? A; from Corrected Q; 9. • Line 89: consider use precision rather than precession? A; precision Corrected Q;10. • Line 124, at the end: consider using annotated rather than noted (famous)? A; annotated Corrected Q; 11. • Sentence in lines 139, 140 and 141: This sentence cannot be understood? A; sentence Corrected Q; 12. • Line 181: consider editing the sentence to change the use of which (". . . obtained by AAS are close to already published data . . .)? A; sentence Corrected Q; 13. • Table 12: Oxygen must be capitalized. (SiO2)? A; O Corrected Q; 14. •Line 301 at the end needs a verb? A; verb Corrected Q; 15. • Line 310 at the end also needs a verb. ? A; verb Corrected Q; 16.• Line 320: It cannot be used the form "there is" paired with "accuracies"? A; word Corrected Q; 17. • Line 342: The sentence at the end needs some verb. • Line 416: elements if (space)? A; verb Corrected Q; 18. • Line 643: background (eliminate space) • Sentence beginning at the end of line 651: the use of "them" is not clear, and verb tenses must be related (past or present)? A; space Corrected Q; 19.• Sentence beginning at line 655 cannot be understood? A; Corrected Q; 20. • The end of sentence in lines 659 and 660 is not clear: high grade of metamorphic rocks? High grade of metamorphism? A; high grade of metamorphic rocks Corrected Q; 21. • Sentence in lines 662 and 663 cannot be understood? A; sentence Corrected Q; 22. Some parts of the text need to be rewritten because they are not clear. Specially the Abstract and the Conclusions are hard to be understood due to the lack of verbs. I encourage the authors to make an exhaustive

revision of all the sections to avoid so much grammar errors? A; Grammatical errors corrected